# Evaluation of a Less Invasive Cochlear Implant Surgery in *OPA1* Mutations Provoking Deafblindness

**DOI:** 10.3390/genes14030627

**Published:** 2023-03-02

**Authors:** Ahmet M. Tekin, Hermine Baelen, Emilie Heuninck, Yıldırım A. Bayazıt, Griet Mertens, Vincent van Rompaey, Paul van de Heyning, Vedat Topsakal

**Affiliations:** 1Department of Otorhinolaryngology Head and Neck Surgery, University Hospital UZ Brussel, Vrije Universiteit Brussel, Brussels Health Campus, 1090 Brussels, Belgium; 2Department of Otolaryngology Head and Neck Surgery, Medipol University Hospital, University of Medipol, Istanbul 34214, Turkey; 3Department of Otorhinolaryngology, Head and Neck Surgery, Antwerp University Hospital, 2650 Edegem, Belgium; 4Department of Translational Neurosciences, Faculty of Medicine and Health Sciences, University of Antwerp, 2610 Antwerp, Belgium

**Keywords:** sensorineural hearing loss, otogenetics, genetic deafness, hereditary hearing impairment, deafblindness, genetic analyses on sensorineural hearing loss, image guided surgery, robot assisted cochlear implantation surgery

## Abstract

Cochlear implantation (CI) for deafblindness may have more impact than for non-syndromic hearing loss. Deafblind patients have a double handicap in a society that is more and more empowered by fast communication. CI is a remedy for deafness, but requires revision surgery every 20 to 25 years, and thus placement should be minimally invasive. Furthermore, failed reimplantation surgery will have more impact on a deafblind person. In this context, we assessed the safety of minimally invasive robotically assisted cochlear implant surgery (RACIS) for the first time in a deafblind patient. Standard pure tone audiometry and speech audiometry were performed in a patient with deafblindness as part of this robotic-assisted CI study before and after surgery. This patient, with an optic atrophy 1 (*OPA1*) (OMIM#165500) mutation consented to RACIS for the second (contralateral) CI. The applicability and safety of RACIS were evaluated as well as her subjective opinion on her disability. RACIS was uneventful with successful surgical and auditory outcomes in this case of deafblindness due to the *OPA1* mutation. RACIS appears to be a safe and beneficial intervention to increase communication skills in the cases of deafblindness due to an *OPA1* mutation. The use of RACIS use should be widespread in deafblindness as it minimizes surgical trauma and possible failures.

## 1. Introduction

Hearing loss has a significant impact on people’s quality of life and daily communication skills in their community. Hearing impairment, which currently affects approximately 360 million people worldwide, is the most common sensory deficit in humans [1]. Hearing loss is, therefore, on the list of priority diseases by the World Health Organization for research into therapeutic interventions to address public health needs [2,3]. It is estimated that 60% of severe sensorineural hearing loss cases are of genetic origin, and genetic hearing losses are divided into syndromic and non-syndromic types [4,5,6]. Patients with syndromic hearing loss involving visual impairments will have major limitations in their lives such as social isolation, which may lead to decreased physical activity and an increased risk of physical harm [7]. Deafblind patients have a dual sensory handicap in a society that is becoming more and more empowered by fast communication. In this context, long-term treatment of deafness with cochlear implantations (CI) seems critical. Although manufacturers provide short warranties for their implants, as a mechanical device, they may have a life expectancy of 20 or even 25 years [8]. Then, revision surgery is usually required to replace the broken or worn-out implant. Although this may seem straightforward, if this fails, it leaves the patient deaf after benefitting from the implant for 20 to 25 years [9]. Since this can be devastating for a 20–25-year-old adult in the beginning of their career or studies, a bilateral implant is generally advised, especially in deafblind patients where there is a dual sensory deficit. Therefore, it is also important to perform the surgery in the least invasive way so that revision surgery will not be compromised. This should be the standard of care for any type of deafness, but the consequences of a failed revision surgery may be more devastating in deafblindness, especially because there are still no vision-restoring implants clinically available that are as efficient as CI. In fact, CIs are the most effective implants restoring a neurosensory deficit in medicine.

### 1.1. Autosomal Dominant Optic Atrophy

Autosomal dominant optic atrophy (ADOA) (OMIM#165500) is one of the most common neuro-optic disorders resulting from optic nerve degeneration with a prevalence of 1 in 50,000 people. In the syndromic forms, ocular problems such as bilateral progressive loss of vision, pallor of the optic disc, central loss of vision, and impaired color vision can be accompanied by non-ocular neurodegenerative problems [10,11]. The optic atrophy 1 (OPA1) gene, which is located on chromosome 3q28-29, is associated with the degeneration process in ADOA patients [10,12]. The OPA1 protein is composed of mitochondrial target signal (MTS), GTPase/dynamin domain, trans-membrane domain (TM), and presenilin-associated rhomboid-like protease (PARL) [13]. The OPA1 protein is known to be responsible for various mechanisms including mitochondrial fusion, oxidative phosphorylation, membrane potential permeability, and apoptosis control [14,15,16]. The GTPase/dynamin domain can play an important role in the interaction of mitochondrial membrane proteins and mitochondrial fusion, and impairment in GTPase activity can lead to uncontrolled proton leak as a result of impaired inner mitochondrial membrane structure and reduced membrane potential [17]. In addition to optic neuropathy, more than 20% of patients with OPA1 mutations manifest neurodegenerative problems such as ptosis, ataxia, peripheral neuropathy, mitochondrial myopathy, and progressive external opthalmoplegia as well as bilateral sensorineural hearing loss [18,19,20,21]. OPA1-related hearing loss is considered to increase the likelihood of being specific for mitochondria-rich auditory nerve fibers [22]. The use of CI in patients carrying the OPA1 mutation improves speech perception and synchronous activity in auditory brainstem pathways by bypassing the site of the lesion [21,23,24,25] (Table 1). It seems of utmost importance in this condition not to harm the residual hearing and perform structure-preserving surgery to the highest standards.

### 1.2. Cochlear Implantation Surgery for Deafblindness c.1499G>A p.(Arg500His)

As mentioned, a CI has a life expectancy of 20 to 25 years [8]. This means that a child may need at least two revision surgeries during their life to allow hearing with an implant. Every surgery may lead to tissue problems such as foreign body reactions, fibrosis, or biofilm formation [26,27,28,29]. Therefore, placement should be minimally invasive for every patient. A failed reimplantation surgery will have more impact on a deafblind person. Consequently, care should be exercised to preserve residual hearing while minimizing surgical trauma in an attempt to decrease the possibility of revisions and device failures in the long-term. The surgeon who performs CI placement in patients with deafblindness should follow the rules of soft surgery [30,31]. Atraumatic surgery facilitates hearing preservation, better CI outcomes, and the protection for future treatment strategies such as the regeneration of hair cells [29,32,33]. Thus, a minimally invasive CI surgery seems important, especially in cases of deafblindness. Some consensus exists on hearing-preserving surgery (e.g., regarding aspects such as round window insertion, slow insertion, and insertion in the right angle with soft electrodes) [34]. Most factors exceed human dexterity and therefore are not yet standardized.

Robotic technology is thought to have the potential to exceed these limits, allowing for minimally invasive cochlear access and a controlled electrode placement [35]. After the introduction of stereotactic frame-based keyhole intervention by Labadie et al. in 2015 [36], details of robotically assisted cochlear implantation surgery (RACIS), robotically assisted drilling for the first time from the cortex of the mastoid through the facial recess into the middle ear cavity, and manual drilling for access to the inner ear and insertion of the electrode were shown by Weber et al. [35] in 2017. Afterward, Topsakal et al. performed inner ear access with robotically assisted drilling for the first time [37] by controlling the angles [38] in the cochlear approach (Figure 1) and reported the importance and necessity of customized surgery for specific genetic conditions such as a POU3F4 mutation, causing an incomplete partition type III (IP-III) [39]. Surgical skill for atraumatic correct placement in CI surgery is related to the angles under which the array is inserted. At this point, RACIS has demonstrated its precision and accuracy regarding the entry angles into inner ear access [38]. RACIS follows a minimally invasive direct access to a designated target, rather than a greater surgical exposure, which is one of the principles of soft surgery [39]. Within this, the application of safe surgical exposure requires the identification of landmarks [38]. RACIS sometimes uses data beyond human perception to guarantee security and accuracy.

Since we know the importance of pre-operative planning, intraoperative imaging, and the use of landmarks in OPA1-related deafblindness, we aimed to demonstrate in this study, for the first time, the application of RACIS as a minimally invasive surgery in these patients in an attempt to minimize surgical trauma. We also compared conventional surgery versus robot-assisted surgery on separate sides of the same patients offered bilateral implantation.

## 2. Materials and Methods

### 2.1. The Patient

A 38-year-old female with an *OPA1* mutation who previously had a CI in her left ear was referred to our tertiary referral center for otology and neurotology for diagnostic work-up and candidacy selection for RACIS on her right ear.

### 2.2. Audiological Evaluation

As part of the CI intake protocol, hearing tests were performed in this deafblind patient both pre- and postoperatively, according to the ‘Minimal Outcome Measurements’ (MOM) procedure [40]. Unaided pure tone audiometry was conducted pre-operatively according to ISO 8253-1 (2010) standards using 5A10 insert earphones and a bone conductor to obtain pure tone air and bone conduction thresholds. Aided thresholds with the CI were measured post-operatively with warble tones in a free field with a loudspeaker at a distance of one meter in front of the listener. Hearing thresholds were determined in the frequency range between 125 Hz and 8 kHz. Speech perception in quiet environments was performed using the Dutch open-set NVA lists [41], consisting of monosyllabic consonant-vowel-consonant (CVC) words, at 65 dB SPL. Speech perception in noisy environments was performed using the Leuven Intelligibility Sentences Test (LIST) with an adaptive procedure as described in van Wieringen [42]. Speech perception in quiet and noisy environments was performed in unaided conditions pre-operatively and post-operatively with the CI. All speech perception tests were performed in a free field with the loudspeaker at 0° azimuth at one meter of the subject.

### 2.3. Molecular Analysis

Molecular analysis was performed on DNA extracted from fresh blood. Variant analysis was performed by Next Generation Sequencing (NGS) on a NextSeq500 sequencer (Illumina, San Diego, CA, USA) after Haloplex enrichment of a gene panel consisting of 99 genes known to be implicated in non-syndromal hearing loss. Sequence data were analyzed with SeqNext analysis software (JSI medical systems, Ettenheim, Germany). For all individual genes, 30× coverage was obtained for more than 95% of the coding sequence, and for the total gene panel, 30× coverage was obtained for more than 98% of the total coding sequences of all genes. A minimal minor allele frequency threshold of 15% was used for variant detection. Potentially pathogenic variants were confirmed by Sanger sequencing. Classification of variants was performed according to the American College of Medical Genetics and Genomics guidelines [43].

### 2.4. RACIS: Robotically Assisted Cochlear Implant Surgery

Written informed consent was obtained from the study case for RACIS, genetic testing, and the anonymous use of data for scientific purposes. We applied a newly developed RACIS for types of deafblindness. The study (ClinicalTrials.gov NCT04102215) was conducted with the approval of the ethics committee of the Antwerp University Hospital (B300201941457). The HEARO system^®^ (Figure 2) (CAScination AG, Bern, Switzerland), a new generation otologic surgical robot based on the Otobot system [35], integrates a set of sensors, actuators, and core functions to enable the surgeon to perform image-guided surgery with a robotic arm.

With dedicated planning software (OTOPLAN^®^, CAScination AG, Bern, Switzerland), the surgeon produced a 3D reconstruction of all relevant anatomical structures and designated the target (Figure 3) to enter the inner ear parallel with the basal turn, through the round window, and to remain as lateral as possible with a flexible electrode [44]. This trajectory accommodated the safety distance to the critical structure while minimizing the in- and out-plane angles. The HEARO procedure is designed for a minimally invasive round window approach. Surgery proceeds by maintaining appropriate distances between the operation orbit of the robot and critical anatomical structures, as can be seen in Figure 3. Here, we wanted to focus on the minimally invasive approach and for further details of the HEARO procedure, we refer to the paper discussing the clinical trial, since the surgical details exceed the scope of this paper [37].

## 3. Results

### 3.1. Molecular Genetic Analysis in OPA1 Mutation and Correlation to Phenotype

A positive familial history of hearing loss was present: the proband of the family (I:1), her daughter (II:3), and her daughter’s daughter (III:1) had deafblindness (Figure 4). Next-generation sequencing of a hearing loss gene panel in her granddaughter showed a heterozygote c.1499G>A p.(Arg500His) likely pathogenic variant in the *OPA1* gene (NM_130837.2). Her daughter (II:2) and her granddaughter (III:2) were heterozygous carriers. Apart from this, no possible variant that could be the cause of hearing loss was observed.

### 3.2. Surgical Results

Full insertion of the electrode array was performed with an uncomplicated surgical procedure. The intraoperative mobile cone beam computed tomography (CBCT) XCAT XL (XORAN, Ann Arbor, MI, USA) demonstrates an ultimate lateral wall positioning of FLEX28 (MEDEL GmbH, Innsbruck, Austria) with a full cochlear coverage of 537 degrees (Figure 5).

In the control computed tomography images taken after the surgery, the conventional cochlear implant on the left side and the RACIS on the right side are seen in the coronal and axial sections (Figure 6). The conventional placement of the CI on the left side was combined with the obliteration of the mastoidectomy with bony dust. A clear alteration in the bony structures and disrupted mastoid air cells can be seen compared to the minimally invasively operated right side.

### 3.3. Audiological Findings

The patient with an *OPA1* mutation presented a bilateral profound sensorineural hearing loss with a PTA of 105 dB HL in the left ear and a PTA of 95 dB HL in the right ear. Hearing loss was detected at the age of 12 years. The patient was first implanted unilaterally at the University Hospital of Antwerp at the age of 36 years. She was implanted with conventional surgery in the left ear. Two years after the first implant, she received a CI in the contralateral ear with RACIS. The first activation of each implant’s speech processor took place 4 weeks after surgery and both CIs were fitted with a standard behavioral fitting method. Since the postoperative controls of the right side, unfortunately, coincided with the COVID-19 pandemic, the control results could not be recorded the first year after implantation due to restrictions. Therefore, the audiological outcomes after two years of CI use are reported for the right side as well as for the left side for comparison. Sound field thresholds, speech perception in the quiet, and speech perception in a noisy environment were improved in both the conventionally implanted and RACIS implanted side. The patient’s audiological outcomes are shown in Table 2.

## 4. Discussion

In this patient with dual sensory loss, the minimally invasive robotic and soft surgery in an individually planned orbit according to the visualization of each patient’s unique anatomy before entering the operating room was critical. Patients with variations in the *OPA 1* mutation have a tendency to lose ganglion cells due to the genetic condition affecting mitochondria and scar formation due to surgery.

It has been stated that the visual and hearing impairments in these patients lead to restrictions in their social lives [7]. Poor physical and psychosocial health may lead to an increase in fatigue and even suicidal behavior in deafblind patients [45]. For this reason, it seems important to treat where possible the deafness and or blindness in the most competent way in order to ensure that deafblind people remain connected to their social life. Interventions should be performed early and under the optimal conditions. In this context, we aimed to restore the auditory loss with a CI at the earliest convenience, aiming to facilitate neural plasticity [46]. One should also not forget that vision and hearing are warranted by two end organs, and bilateral therapy should not be overlooked. Although this may seem straightforward, in some countries, insurance companies only reimburse one cochlear implant [47]. The costs of implants restoring vision are perhaps even higher. According to previous studies, a CI is an effective treatment in cases of deafblind patients with Usher syndrome (OMIM#276900) in terms of quality of life, verbal communication skills, and cognitive function [48,49,50,51,52]. The reports of the beneficial effects of CIs have also changed the practice of reimbursement, and a second CI could be reimbursed in adults in extraordinary cases. In this context, most Dutch insurance companies have been reimbursing a second CI in cases of meningitis. Similarly, the Netherlands and Belgium have reimbursed a second CI in cases of deafblindness.

In patients with the *OPA1* mutation, despite the presence of OPA1 protein in the inner and outer hair cells, auditory nerve endings, and spiral ganglion cells, it is still unclear which cochlear region is most vulnerable to degeneration [13,22,23]. However, it is evident that the sensorineural hearing loss looks like an auditory neuropathy as the otoacoustic emissions are preserved while the auditory brainstem responses are abnormal [13,23,53]. CI outcomes of the patients with auditory neuropathy can differ depending on the site of the underlying mutation [54]. CI seems beneficial in cases of OTOF mutation and Brown–Vialetto–Van Laere syndrome-related auditory neuropathies (OMIM#211530) [55,56], whereas patients will benefit less from a CI in cases of mutations affecting the auditory ganglion cells and auditory nerve [57,58]. Therefore, scar formation and a loss of neuronal structure at the neural ganglion, which are not well-described. We have anecdotal evidence that minimally invasive and custom surgeries or tailored surgeries are necessary, and future studies will do well to focus on this. CIs are also beneficial in *OPA1* mutations in terms of the restoration of hearing thresholds, speech perception, and synchronal activity in the auditory brainstem [21,23,24,25]. Similarly, our patient with an *OPA1* mutation benefited from her CIs. Although the association of the *OPA1* mutation with hearing loss has been reported in numerous studies [11,18,19,20,21,22,23], there are only a reported 11 patients with ADOA syndrome (OMIM #125250) whose hearing was affected due to the *OPA1*-related mutation and who received a CI [21,23,24,25] (Table 1).

The surgical technique must be adapted to the genetic condition for hearing and structure preservations, and obviously, less traumatic electrodes are indicated. In addition, straight lateral wall electrodes will harm the nerve cells even less, which are endangered in this condition [44]. This genetic condition and how it affects the nerve endings must be investigated. Robotics have been important technological breakthroughs in CI surgeries. Safe surgical techniques for approaching the cochlea in image-based RACIS and the proper distances between the operation orbit of the robot and critical anatomic structures have been reported [38]. RACIS has been shown to cause less trauma to the cadaveric cochlea compared to manual insertion [59]. In addition, RACIS is minimally invasive because of a direct keyhole trajectory toward the inner ear spearing the healthy mastoid, and it involves less noise-induced trauma from the surgical drill. The direct trajectory parallel with the scala tympani [60] will allow a soft electrode to be inserted according to the principles of soft surgery. Herein, the safe application of a new-generation RACIS procedure was presented in patients with ADOA syndrome.

RACIS enables us to minimize surgical limitations of the surgeon for repeatable, minimally invasive cochlear access, and thereby increase the number of eligible patients for CI. Reference recording is an important step of the robotic procedure due to mastoid thickness being an important anatomic parameter, and reference implantation should be performed before planning the surgery [61]. In our case, we saw that the current anatomic status of the *OPA1* mutations could allow us to plan and perform a correct and safe robotic orbit and insert the electrode. As the anatomical variations would be more common in syndromic patients, reference implantation should be carried out more carefully in these patients, we should increase our experience through performing RACIS in a larger number of syndromic patients. In our previous study, we reported the first case of IP-III, one of the rare inner ear anomalies, in which RACIS was successfully performed [39]. We argue that with a newly developed system of robotically assisted and image-guided approaches and facial nerve monitoring, the idea of robotic surgery has pushed everyone to develop a complete set of new technologies. This is a turning point because we can now carry this out without any complications in a syndromic patient with dual sensory loss with precision. Surgeons should embrace this technology to standardize surgical outcomes in syndromic and potentially difficult cases in order to serve their patients.

Behr’s syndrome (OMIM #210000), the most severe condition caused by *OPA1* gene mutations, is accompanied by neurological issues that start in early life. Behr’s syndrome causes ocular atrophy, encephalopathy, peripheral neuropathy, loss of sensation and muscle weakness in the limbs, ataxia, feeding and digestion issues, and developmental delays in affected individuals. The characteristics of these illnesses are probably brought on by the loss of cells in numerous tissues as a result of deficient mitochondrial function [19,62]. Why *OPA1* gene mutations only impact the eyes in people with optic atrophy type 1 but have more significant effects in ADOA-plus syndrome is unknown. Researchers hypothesize that some *OPA1* gene mutations result in the development of an altered protein that interferes with the function of the normal protein produced from the non-mutated copy of the gene, further affecting the function of the OPA1 protein. Optic atrophy type 1 and ADOA-plus syndrome are caused by mutations in the *OPA1* gene in one copy of the gene in each cell, but Behr’s syndrome is caused by mutations in both copies of the *OPA1* gene in each cell. Mutations in both copies of the gene significantly limit the quantity of effective OPA1 protein, which most likely contributes to the severe Behr’s syndrome symptoms and manifestations [63,64,65,66,67]. The widely varied clinical manifestation seen both between and within families harboring the same variation, in general, makes genotype–phenotype correlations in OPA1-associated disorders difficult to establish [68]. In fact, the prevalence may vary greatly, ranging from 43 to 88% [69,70,71], indicating the existence of as-yet undescribed modifying variables that may be able to affect how the ADOA-plus syndrome manifests phenotypically. Sensorineural hearing impairment was found in 62.5% of *OPA1* mutation carriers and was the second most common main clinical characteristic in ADOA-plus patients among the extraocular diseases, other than optic neuropathy brought on by *OPA1* gene mutations. These findings imply that mitochondrial fusion impairment and neuronal death are the cause of the deafness associated with this *OPA1* mutation [72]. Hearing was restored via cochlea electrical stimulation of the auditory nerve’s proximal myelinated parts [23].

Surgeons may have reservations regarding the possibility of complications or revisions, which can be catastrophic in patients with dual sensory loss. At this point, robotic surgery can be helpful. In RACIS, we can evaluate the patient-specific anatomy that will minimize the margin of error before the operation and apply soft surgery during the electrode placement.

## 5. Conclusions

This manuscript describes, for the first time, that RACIS appears to be a safe and beneficial intervention to increase hearing and communication skills in cases of deafblindness due to *OPA1* mutations. *OPA1* mutations causing deafblindness require even more attention to be paid to structure and tissue preservation during CI, and the use of RACIS can be helpful. For the moment, RACIS has only proven its inferiority on audiological outcomes, but surgically, it surely outperforms human dexterity in terms of accuracy and consistency. RACIS is paving the way to standardizing the surgical outcomes to the maximum.

## Figures and Tables

**Figure 1 genes-14-00627-f001:**
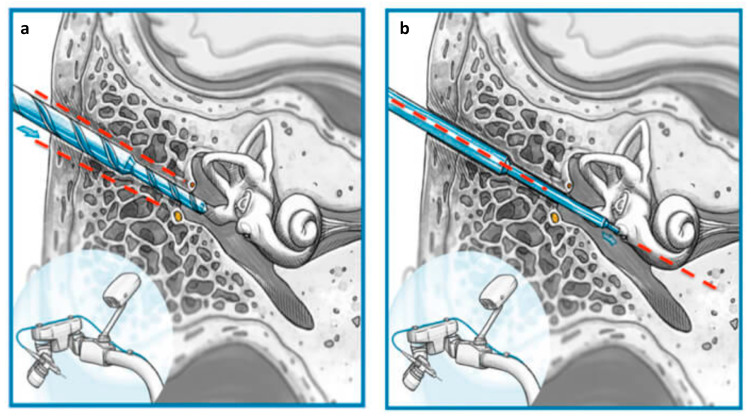
(**a**) Illustrates the middle ear access performed. (**b**) Illustrates the inner ear access performed.

**Figure 2 genes-14-00627-f002:**
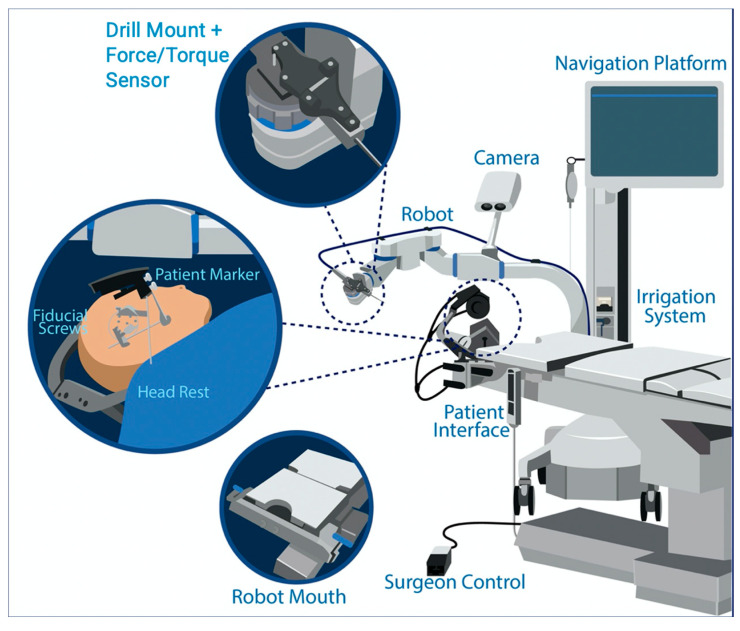
The HEARO system.

**Figure 3 genes-14-00627-f003:**
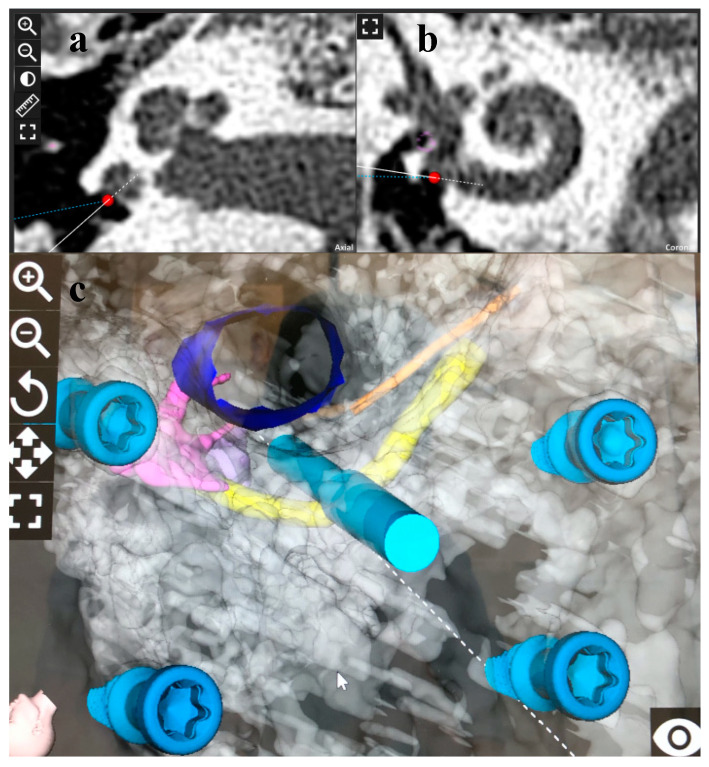
(**a**,**b**) Pre-op cone beam computed tomography axial and coronal views showing the target plan. (**c**) Pre-op planning with 3D reconstruction of all relevant anatomical structures; yellow color showing facial nerve, orange color showing chorda tympani, dark blue color showing external auditor canal, pink color showing incus and malleus, purple color showing stapes.

**Figure 4 genes-14-00627-f004:**
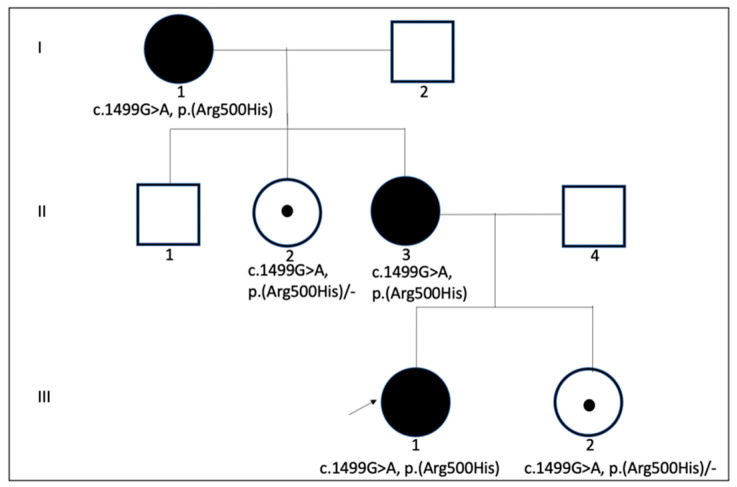
The three-generation family pedigree demonstrating segregation pattern of OPA1 c.1499G>A p.(Arg500His). Roman numerals indicate generations. The arrow indicates the case that underwent robotic surgery.

**Figure 5 genes-14-00627-f005:**
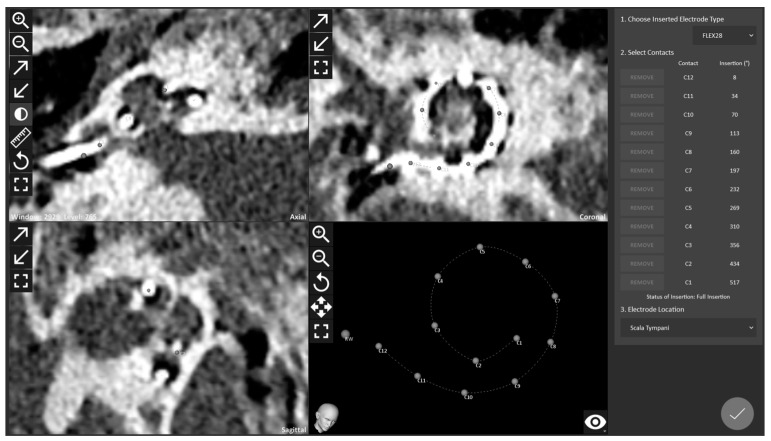
Post-op CBCT showing all fully inserted 12 electrodes in the cochlea.

**Figure 6 genes-14-00627-f006:**
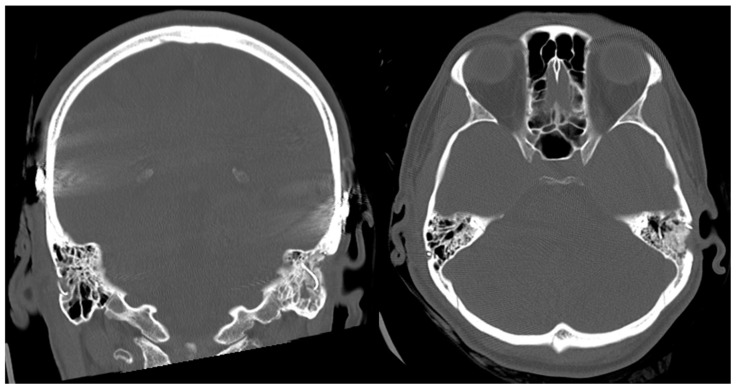
Coronal and axial sections of a CT scan of the post-operative period.

**Table 1 genes-14-00627-t001:** Mutations in the *OPA 1* gene reported in the literature in patients who underwent cochlear implantation.

Nucleotide Change	Amino Acid Change	Feature of Deafness	Cochlear Implantation (CI)	Improvements in Auditory and Speech Performance after CI	Location	Year	References
c.1316 G>T	p.G439V	SNHL	Unilateral	No	Italy	2015	[21]
c.1316 G>T	p.G439V	SNHL	Unilateral	Yes	Italy	2015	[21]
c.869 G>A	p.R290Q	SNHL	Unilateral	Yes	Italy	2015	[21]
c.1334 G>A	p.R445H	SNHL	Unilateral	Yes	Italy	2015	[21]
c.1334 G>A	p.R445H	SNHL	Unilateral	Yes	Italy	2015	[21]
c.893 G>A	p.S298N	SNHL	Unilateral	Yes	Italy	2015	[21]
c.1334 G>A	p.R445H	SNHL	Unilateral	Yes	USA	2015	[21,23]
c.1334 G>A	p.R445H	SNHL	Unilateral	Yes	USA	2015	[21,23]
c.892A>C	p.Ser298Arg	SNHL	Unilateral	Yes	Japan	2019	[25]
c.1334G>A	p.Arg445His	SNHL	Unilateral	Yes	Japan	2019	[25]
c.1414T>C	p.Cys472Arg	SNHL	Bilateral	Yes	Taiwan	2022	[24]
c.1499G>A	p.(Arg500His)	SNHL	Bilateral	Yes	Belgium	2022	Present Study

**Table 2 genes-14-00627-t002:** The subject’s audiological outcomes.

Implanted Ear	Type of Surgery	Age at Implantation (Years)	Inactive Electrodes	PTA_0.5; 1; 2 and 4 kHz_ (in dB HL)		Speech in Quiet at 65 dB SPL (% Correct)		Speech in Noise (dB SNR)	
				Pre-op Unaided	Post-op CI Ear (2 Years)	Pre-op Unaided	Post-op CI Ear (2 Years)	Pre-op Unaided	Post-op CI Ear (2 Years)
Left	Conventional	36	/	105	40	0	91	>20	+2
Right	RACIS	38	/	95	30	0	88	> 20	+0

## Data Availability

The data presented in this study are available in the article.

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
