# Peer review of "Evaluation of a Less Invasive Cochlear Implant Surgery in OPA1 Mutations Provoking Deafblindness"

_genes, 2023, doi:10.3390/genes14030627_

Round 1
Reviewer 1 Report
I have reviewed with pleasure the manuscript titled: "Evaluation of a Less Invasive Cochlear Implant Surgery in OPA1 Mutations Provoking Deafblindness". It is very important to preserve residual hearing and to perform the procedure with minimal trauma,This becomes even more important when there are no solid landmarks in people with different anatomical structures.
Deafblindness cases attributed to OPA 1 mutation will hopefully pose less surgical challenges with technological advances. In today's communication times, cochlear implant surgery for deafblind patients is important. Revision of the applied cochlear implant may be required in these patients. Therefore, minimally invasive surgery is important in the initial surgery. The successful application of the robotic cochlear implant with minimally invasive surgery in these patients is demonstrated. The robotized approach to the cochlea for cochlear implantation will be with no doubt the future of cochlear implant surgery. It should allow for mini-invasive surgery with minimal and safe drilling of the mastoid bone, assisted by CT-scan-guided robotic surgery.
The article comprises relevant information. The use of CI in patients with the OPA1 mutation improves speech perception and synchronous activity in auditory brainstem pathways.
At the same time, this genetic hearing loss accompanies blindness, it is very important for the surgeon to decide to operate and to perform this surgery. Cochlear implant surgery applied to these patients is limited in the literature. In this case, it seems extremely important to perform minimally invasive surgery at the highest standards and tissue preservation during cochlear implantation. Standardization of surgical results with robotic cochlear implant surgery, as well as providing a patient-specific placement trajectory, will open the way for more widespread surgery in these patients with OPA 1 mutation.
Cochlear implantation has previously been demonstrated in limited patients with hearing loss due to the OPA 1 mutation. However, it has been shown for the first time that this surgery is successfully performed in these patients with robotic surgery and its results are similar to conventional surgery
minor comments:
Comparing hearing outcomes between robotic and conventional cochlear implant surgery would be more appropriate if more patient series were included in the comparison.
Please provide information whether information on variants displayed in table 1 may be found in public databases, e.g.,clinvar
I congratulate the authors for using technology well on this subject.
Reviewer 2 Report
The study aims to demonstrate the feasibility of robotic surgery to minimize the surgical trauma in cochlear implant surgery in deaf blind subjects.
Round window insertion, slow insertion and insertion in the right angle with soft electrodes can be more precise and soft with the robot but there is no manual feedback of possible obstacles
Patient preparation will be longer than operation itself
Minimal mistake in positioning of navigation tools can became a source of catastrophic complications
Revision surgery to change the implant after 20-25 years will be extremely difficult with so limited approach in the first intervention
The Authors should prove that robotic surgery is safer than manual surgery by reporting data on early and late hearing preservation.
There are some typing mistakes along the manuscript
Reviewer 3 Report
The authors describe the use of robotic CI surgery in a patient with OPA1 gene mutation or optic atrophy type 1 and deafness. The deafness is believed by the authors to be caused by a degeneration of the unmyelinated peripheral fibers of the auditory nerve. Robotic surgery may improve the correctness of electrode insertion angle and reduce traumaticity. It may lead to greater preservation of tissue and be important at future reimplantation. The paper is therefore of high value especially in this category of patients. Images are very good.
However, the paper is not so well written and should be scrutinized by a language editor. There are many errors in spelling and the figure captions should be corrected. Many sentences are unnecessarily long and grammar should be improved. The peripheral nerve fibers are not unmyelinated as stated. Maybe the authors mean the synaptic nerve terminals at the inner hair cell bodies. If so it should be explained. The abbreviations such as OPA1 should be explained initially to the reader. I guess Behr syndrom should be Behr´s syndrom.
Round 2
Reviewer 2 Report
The patient of this report was already included in a previous paper on the same issue. I don't understand why there is a need to propose again the same arguments in another article.
